# Bond Strength Stability of Different Dual-Curing Adhesive Cements towards CAD-CAM Resin Nanoceramic: An In Vitro Study

Edoardo Alberto Vergano [1], Andrea Baldi [1], Allegra Comba [1], Edoardo Italia [1], Giorgio Ferrero [1], Rossella Femiano [2], Felice Femiano [2] and Nicola Scotti [1,*]

1    Department of Surgical Sciences, University of Turin, 10133 Turin, Italy; edoardo.vergano@gmail.com (V.E.A.); andrea.baldi@unito.it (B.A.); allegra.comba@unito.it (C.A.); italiaedoardo@gmail.com (I.E.); giors95@gmail.com (F.G.)
2    Restorative Dentistry, University of Study of Campania, 81100 Naples, Italy; rossella.femiano@libero.it (F.R.); femiano@libero.it (F.F.)
*    Correspondence: nicola.scotti@unito.it; Tel.: +39-3402861799

**Abstract:** Background: To evaluate different adhesive luting procedures on coronal dentin bond-strength of Cerasmart CAD-CAM blocks with µTBS test. Methods: 36 molar crowns were flattened in order to expose sound dentin and a standardized smear layer was created with 600 grit paper. Specimens were divided into six groups according to the luting cement employed (n = 12 each): G1: Panavia V5 (Kuraray, Japan); G2: Bifix QM (Voco, Germany); G3: Estecem (Tokuyama, Japan). CAD-CAM blocks (Cerasmart, GC), shade A2LT, size 14, were sectioned with a diamond saw to obtain 4 mm high specimens, which were then luted on the coronal dentin, following the manufacturer instructions. Specimens were serially sectioned to obtain 1 mm thick beams in accordance with the µTBS test technique. Half of the beams were stressed to failure after 24 h (t = 0), while the other half were stored in artificial saliva for 12 months, at 37 °C, for ageing before stressing to failure (t = 12). Results: two-way ANOVA test showed significant difference for the factor "luting cement" ($p$ = 0.0002), while the factor "time of storage" ($p$ = 0.0991) had no significant effect on µTBS. Conclusions: PanaviaV5 seems to have better µTBS values at T0 than QM and ES and 1 year aging doesn't seem to affect the bonding strength of tested systems.

**Keywords:** CAD-CAM material; dual-curing cement; bond strength; aging

## 1. Introduction

Machinable materials are nowadays widely used in restorative dentistry and computer-aided design/computer-aided manufacturing (CAD-CAM) approach has become the only way to produce restorations of certain materials, such as cubic zirconia. As stated in the classification by Gracis [1], all restorative ceramic materials can be classified into three big families: glass matrix ceramics, polycrystalline ceramics and resin matrix ceramics. Resin nanoceramics (e.g., Lava Ultimate), glass ceramic in a resin interpentrating matrix (e.g., Enamic) and Zirconia-silica ceramic in a resin interpenetrating matrix (e.g., Shofu Block HC) belong to the last group. Cerasmart (GC Europe, Leuven, Belgium) can be classified as a family near resin nanoceramics, since it consists of evenly dispersed ceramic nanoparticles in a high density composite (71% of silica and barium glass filler by weight) [2].

These materials, with a prevalent resin composition, have several functional advantages: their mechanical properties and their "cushion effect" can lead to a reduction of occlusal forces and, therefore, restoration fracture or antagonist abrasion [3]. Besides these advantages, their composition leads to lower light transmission, causing a diminished curing reaction. A proposed way to avoid polymerization problems is the initial reconstruction of the tooth with an adhesive build up in order to decrease the reconstruction's thickness, associated with the use of a dual curing adhesive/cement [3].

According to the type of curing, adhesive cements can be classified as: chemical cured (CC), light-cured (LC) and dual cured (DC). CC cements are indicated for thick restorations, radicular posts and crowns made of light-blocking materials [4], but their reduced working time and their low esthetical properties limit their use [5]. On the other hand LC cements are the best choice for situations where light can be almost totally transmitted through the restoration, such as translucent veneers and shallow inlays [6]. In situations where light is highly attenuated, such as indirect posterior restorations, DC cements are the best choice, granting controlled working time and short setting time [7]. A proper light activation, after 1 min of exclusive chemical cure, is however crucial in giving the DC cement a proper polymerization [8].

Independently from the curing kinetics, cements can require or not the use of adhesive systems. On one hand, self-adhesive cements simplify the adhesive luting procedure thanks to a chemical reaction like self-etching adhesives. The acidity of these cements is gradually neutralized when reacting with apatite, causing a shift from a hydrophilic compound to a more hydrophobic one [9]. Although they find clinical application with fiber posts, giving the best retention, their mechanical properties and wear resistance are poorer than conventional resin cements [10].

On the other hand, conventional resin cements still rely on the fundamental principles of adhesion with two main approaches: etch-and-rinse (ER) or self-etch (SE). These systems base their effectiveness on different interactions with the dental substrate. The ER approach is based on the complete removal of the smear layer created during cavity preparation, in order to micromechanically interlock with enamel and dentin, while SE employs a smear layer as a substrate for chemical and mechanical adhesion [11,12]. In both cases the interdiffusion zone between collagen and resin, called the hybrid layer, is present, granting the possibility of copolymerization with composite resins, creating a durable bond.

A new class of adhesive systems, called "universal" or "multimode" was introduced with the possibility of being used both in SE and ER mode [13]. However, different works in literature showed that, despite good results on dentin, the etching procedure provides better µTBS results on enamel [14]. Some of these new universal adhesives hold in their formulation a silane coupling agent that helps create siloxane bonds between different materials. It has been demonstrated that this molecule has a precise and short duration effect, and in order to achieve better adhesion towards CAD-CAM silicate materials, it's favorable to add a fresh silane before bonding procedures [15].

The durability of CAD-CAM adhesively luted restorations is not only affected by the type of cement and the bonding procedures but is also dependent on the stability of the bond over time, and is fundamental to achieving long-lasting restorative treatments. However, few studies focus on the bond strength maintenance of CAD-CAM resin nanoceramics luted over dentin through different luting procedures. Thus, the aim of this study was to evaluate the µTBS of three different adhesive cements on coronal dentin when luted to CAD-CAM resin nanoceramics at baseline (T0) and after 1 year of aging in artificial saliva (T = 12).

The first null hypothesis is that there is no difference in bond strength between different luting cements; the second null hypothesis is that aging time does not influence bond strength over dentin.

## 2. Materials and Methods

Thirty-six intact molars, extracted for periodontal reasons, were selected. The occlusal surfaces were flattened using a low-speed diamond saw (Micromet, Remet s.a.s, Casalecchio di Reno, Italy) to expose sound coronal dentin. A standardized smear layer was created using a 600-grit paper (Hermes Schleifmittel GmbH, Hamburg, Germany). Samples were then divided into three groups (*n* = 12 each) according to the dual-curing adhesive cement employed: Group 1, Panavia V5 (V5, Kuraray); Group 2, Bifix QM (QM, VOCO GmbH); Group 3, Estecem (ES, Tokuyama). A detailed composition of the employed materials is showed in Table 1, while a schematic representation of sample preparation is

shown in Figure 1.

**Table 1.** Materials employed in the study.

| Materials | Label | Company | Category | Composition |
|---|---|---|---|---|
| CAD CAM Blocks | Cerasmart | GC Europe, Leuven, Belgium | Resin-based composite | Filler (71% wt): silica (20 nm) and barium-glass (300 nm) nanoparticles |
| Luting Agents | Panavia V5 Cement | Kuraray Europe GmbH, Hattersheim am Main, Germany | Dual-cured adhesive composite cement (automix syringe) | Paste A: Bis-GMA, TEGDMA, Hydrophobic aromatic dimethacrylate, Hydrophilic aliphatic dimethacrylate, Initiators, Accelerators, Silanated barium glass filler, Silanated, fluoroalminosilicate glass filler, Colloidal silica. Paste B: Bis-GMA, Hydrophobic aromatic dimethacrylate, Hydrophilic aliphatic dimethacrylate, Silanated barium glass filler, Silanated aluminum oxide filler, Accelerators, DI-Camphorquinone, Pigments |
| | Clearfil Ceramic Primer Plus | Kuraray Europe GmbH, Hattersheim am Main, Germany | Silane Coupling Agent | 3-trimethoxylsilylpropyl methacrylate, MDP, Ethanol |
| | Panavia Tooth Primer | Kuraray Europe GmbH, Hattersheim am Main, Germany | Self-etching tooth primer | MDP, HEMA, Hydrophilic aliphatic dimethacrylate, Accelerators, Water |
| | Clearfil Universal Bond Quick | Kuraray Europe GmbH, Hattersheim am Main, Germany | Universal Adhesive | HEMA, bisphenol A diglycidyl-methacrylate (BIS-GMA), 10-methacryloyoxydecyl dihydrogen phosphate (MDP), hydrophilic amide monomers, colloidal silica, silane, sodium fluoride, and CQ |

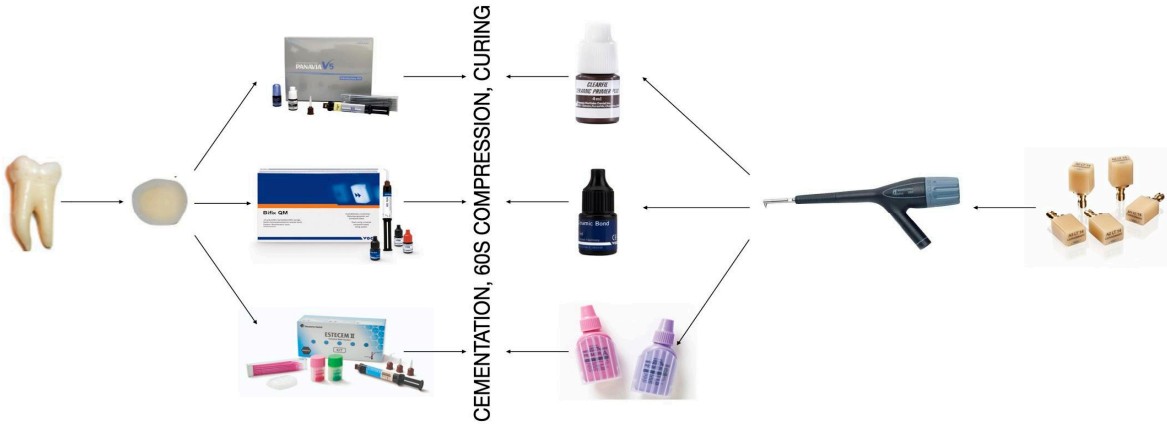

**Figure 1.** Schematic representation of the sample preparation protocol.

Cerasmart CAD-CAM blocks were serially sectioned into 4 mm slabs using a low-speed diamond saw (Micromet, Remet s.a.s, Casalecchio di Reno, Italy). Each slab was then sandblasted with 50 μm $Al_2O_3$ powder at 1.5 bar pressure (Rondoflex, KaVo Italia, Sesto San Giovanni, Italy). Each specimen was then rinsed with white pure alcohol using a microbrush (Microbrush International, Grafton, MA, USA). According to manufacturer instructions and dual-curing cement employed, CAD-CAM slabs were divided into different groups and treated as follows:

Group 1 (G1): Clearfil Ceramic Primer (Kuraray) applied for 20 s with a microbrush and gently air dried.

Group 1 (G2): Bifix Ceramic Bond (VOCO GmbH) applied for 20 s with a microbrush, and gently air dried.

Group 1 (G3): Primer A + Primer B mixed before application in equal proportions (1:1) (Tokuyama), applied for 20 s with a microbrush and gently air dried after 10 s.

Tooth specimens were also randomly divided in different groups (*n* = 12 each) and the exposed dentin area was treated according to manufacturer instructions:

(G1) V5: V5 tooth primer application on the dentinal surface for 20 s, followed by gentle air dry. Application of Clearfil Universal Bond Quick dualized with a 1:1 ratio of Clearfil DC Activator (Kuraray) for 20 s and then gently air dried.

(G2) QM: Application of Futurabond U (VOCO GmbH) for 20 s, then gently air dried.

(G3) ES: Estelink Bond A + Estelink Bond B mixed before application in equal proportions (1:1) (Tokuyama) application for 20 s, then gently air dried.

Adhesively treated specimens were then luted to the pretreated CAD-CAM slabs according to the group assigned: G1: Panavia V5 cement (Kuraray); G2: Bifix QM (VOCO GmbH); G3: Estecem (Tokuyama). After the application of the adhesive cement a constant pressure was applied for 60 s on the composite block to simulate clinical conditions. The specimens were then light cured using a multi-LED lamp (Valo, Ultradent) for 60 s.

After 7 days of storage at 37 °C in distilled water, specimens were serially sectioned, using a low speed diamond saw (Micromet, Remet s.a.s, Casalecchio di Reno, Italy), in order to obtain 1 × 1 mm beams according to the μTBS technique [16]. From each group half of the beams were tested to failure immediately (T0), the other half were stored in artificial saliva prepared according to Pashley et al. 2004 [17], at a 37 °C temperature in an incubator, in order to be tested after 12 months of storage (T12). The beams were stuck on a Ciucchi's jig [18] that doesn't permit bending forces and facilitates the exclusive stressing of adhesion surfaces [19]. A fast-setting cyanoacrylate glue with its activator (Akfix 705, Akkim, Pesar, Italy) was employed for fixating beams. Testing was performed using a microtensile tester (BISCO Dental Products, Schaumburg, IL, USA) at a rate of approximately 1 mm/min. The obtained values, measured in newton (N), were then converted in MPa after measuring the adhesion surface of each stick with a digital caliper.

The number of prematurely debonded sticks in each test group was recorded, but these values were not included in the statistical analysis because all premature failures occurred during the cutting procedure: they did not exceed the 3% of the total number of tested specimens and were similarly distributed within the groups.

A single observer evaluated the failure modes under a stereomicroscope (Stemi 2000-C; Carl Zeiss Jena GmbH) at magnifications up to 50× and classified them as adhesive (A), cohesive in dentin (CD), cohesive in cement (CC), or mixed (M) failures. Representative failures were observed under Scanning Electron Microscope. Different magnification (66×; 150×; 500×; 1000×) images were obtained with the following settings: WD = 10 mm, aperture size = 30.00 μm, EHT = 5.00 kV, signal A = In Lens, stage at T = 0°.

Data were statistically analyzed with a two-way ANOVA to investigate the effect of the factor "luting cement", the factor "time of storage" and their interactions on microtensile bond strength. The Tukey test was used as a post hoc. The Chi-square test was used to analyze differences in the failure modes. Post-hoc pairwise comparison was performed using Tukey test. All statistical analyses were performed using a software (STATA 12, ver. 12.0; StataCorp, College Station, TX, USA) and differences were considered significant for $p < 0.05$.

## 3. Results

Means and standard deviations for μTBS and fractures analysis are showed in Table 2. Results of the two-way ANOVA test showed significant difference for the factor "luting cement" ($p = 0.0002$), while the factor "time of storage" ($p = 0.0991$) had no significant effect on μTBS.

**Table 2.** Microtensile bond strengths (mean ± SD, in MPa) of group 1 (V5), group 2 (QM), group 3 (ES), immediately after bonding (T0) and after 1 year of ageing (T12) in artificial saliva. Number of the failures are reported in parentheses and classified as adhesive (A), cohesive in dentin (CD), cohesive in cement (CC), or mixed (M) failures.

|  | Panavia V5 | Bifix QM | Estelite |
|---|---|---|---|
| T = 0 | 32.45 ± 7.71 (42 A, 1 CC, 2 CD, 10 M) | 25.75 ± 7.65 (36 A, 5 CC, 4 CD, 8 M) | 26.39 ± 6.69 (37 A, 3 CC, 2 CD, 5 M) |
| T = 12 | 35.55 ± 6.18 (34 A, 5 CC, 1 CD, 7 M) | 27.75 ± 6.46 (27 A, 6 CC, 1 CD, 8 M) | 26.32 ± 6.09 (29 A, 10 CC, 4 CD, 9 M) |

Tukey post-hoc test showed only significant difference between Panavia V5 and Bifix QM, with the first achieving better results compared to the second.

A prevalence of adhesive fractures between luting cement and Cerasmart among all the groups was observed at T0 as well as after 1 year of storage in artificial saliva. Representative SEM images are shown in Figures 2–4.

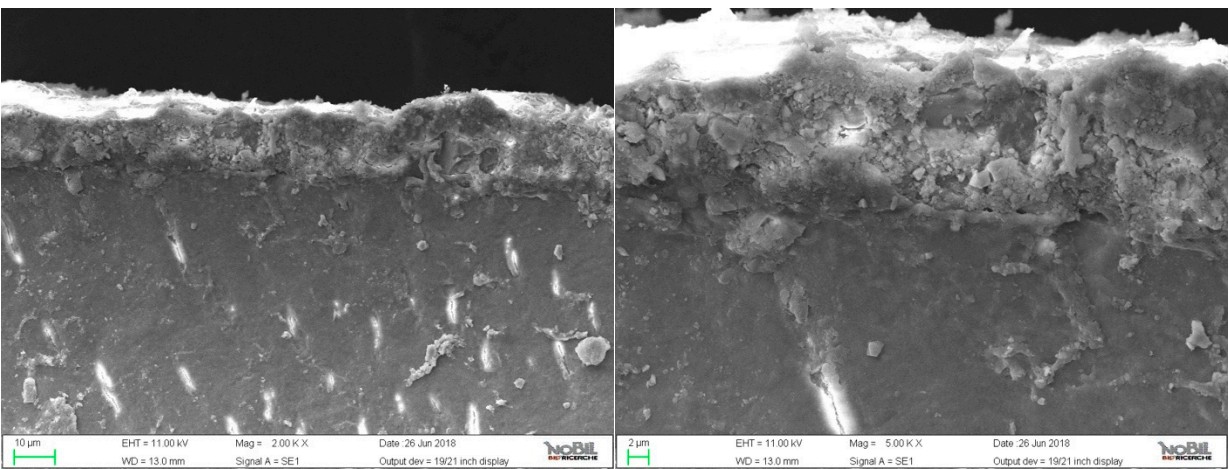

**Figure 2.** Adhesive failure observed at T0: the luting cement (BifixQM) is still bonded to the coronal dentin.

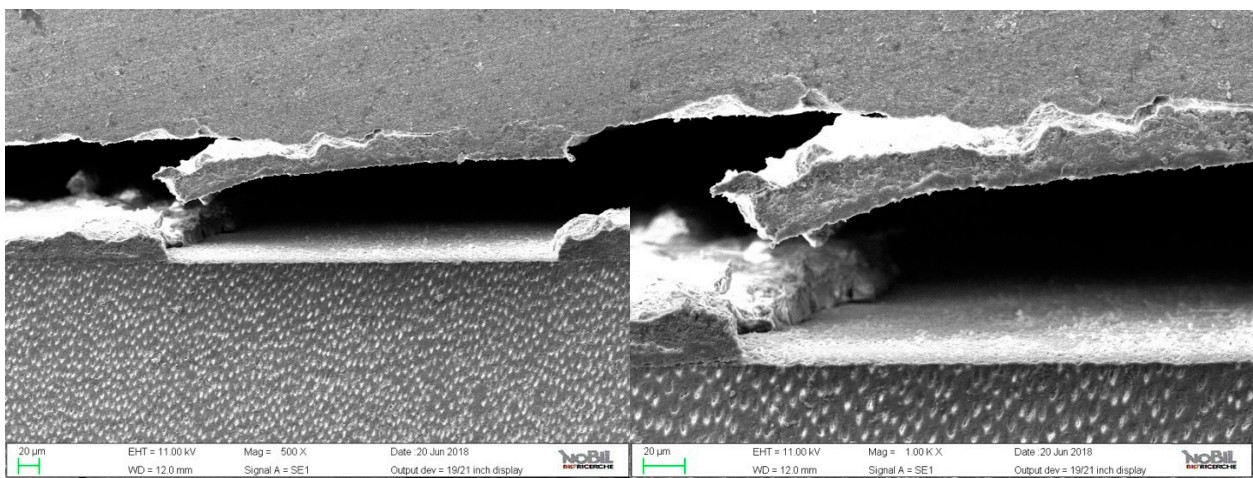

**Figure 3.** A mixed failure, observed in a specimen treated with Panavia V5 and stored in artificial saliva for 1 year.

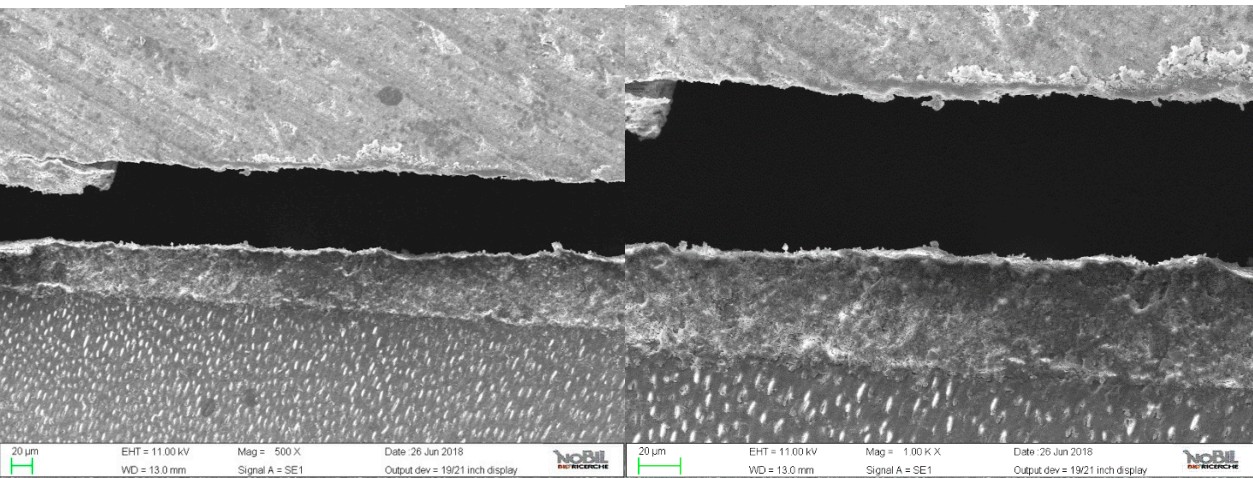

**Figure 4.** A mixed failure, with a prevalence in adhesive pattern, observed in a specimen treated with Estecem and stored in artificial saliva for 1 year.

## 4. Discussion

According to collected data, the first null hypothesis was rejected: results showed that the use of Panavia V5 cement led to higher μTBS values at T0 when compared to Bifix QM and Estecem. The second null hypothesis, on the other hand, was accepted, as aging time did not significantly influence bond strength over dentin.

Many different tests were conducted to analyze bond strength of luting cements: tensile bond strength (TBS), shear bond strength (SBS), and μTBS. According to literature, both the TBS and the SBS test were not considered in the present study as they are less indicated for evaluating adhesive interface: the TBS could be affected by the specimen geometry and the unequal distribution of the interfacial stress [20], while the SBS is supposed to stress more the material itself rather than the hybrid layer, supported by a more frequent cohesive failure pattern observed in SBS samples [21,22]. As stated by Sano et al. [16] μTBS is the most reliable for testing such interface, giving therefore more frequently adhesive failures: this is particularly true when using the Ciucchi's jig for fixating beams to be tested, in order to facilitate manipulation and granting more reliable results [18]. In the present study, in accordance with previous μTBS studies, a prevalence of adhesive fractures was obtained, proving that the adhesive interface was the area where tensile stresses were concentrated.

Manufacturers always suggest a combined use of a particular adhesive system to their luting cement, stating better clinical performances. The three luting systems selected for the present study were tested following these guidelines. Results showed that Panavia V5 had a significantly higher μTBS values at T0 when compared to QM and ES. These results could be explained by the different adhesive protocol and luting cement formulation. The purity of 10-MDP functional monomer composing adhesive and luting systems could lead to different chemical reactions with tooth hydroxyapatite, creating strong insoluble calcium salts [23]. Its highly stable bonding potential could correlate to three reported mechanisms: (1) micro-mechanical interlocking achieved through 10-MDP's etching capacity, and (2) ionic interaction with HAp along with (3) stable monomer-Ca salt formation (nanolayering) [24].

As demonstrated by Yoshihara et al. in 2015 [25], 10-MDP produced by Kuraray showed better bond strength values due to its chemical composition as well as its lower pH, making it capable of a stronger demineralization of dentin itself. On the other hand, another possible explanation may be the presence of chemical kinetics and the use of a separate and fresh DC activator in the adhesive that could increase the effectiveness of Panavia V5 group. The DC activator used for the Panavia V5 method was probably more influential on chemical polymerization compared with the one used in the other cements

tested, because, according to literature, this effect is material-dependent [26]. Adhesive dualization grants the system a first polymerization, with a chemical mechanism, that is then completed by light curing the adhesive and the cement.

Lührs et al. showed that bond strength is statistically influenced by DC and the highest bond strength was obtained when adhesive and luting cement were separately cured [27]. The chemical activation of the adhesive, as performed in the present study, and a chemical conversion of the dual-cure luting cement, followed by photoactivation, probably led to a higher conversion degree of the adhesive-cement complex, which can contribute to justifying the higher bond strength obtained with Panavia V5. This is corroborated by Inokoshi et al. [28] (initial curing characteristics of composite cements under ceramic restorations), who confirmed that the conversion degree is mainly dependent on the composition of the luting cement itself, with Panavia V5 showing a statistically higher conversion rate than other cements therefore possibly leading to higher bond strength values. Therefore, higher conversion values led to significantly higher mechanical properties and, in particular, to higher µTBS [29].

As previously stated, the hybrid layer is crucial for long term adhesive restoration durability. In order to investigate its stability over time, many different ageing protocols were formulated in literature, showing various effectiveness and different time consumption [30]. Since there is no evidence of a best protocol, a standardized approach was used: beams were aged into artificial saliva at 37 °C for 12 months, mimicking oral ambient and therefore its chemical action on adhesive interfaces [31,32]. In fact, it is widely known that the incomplete resin penetration within the hybrid layer leaves gaps, resulting in unprotected type I collagen fibers: this phenomenon takes the name of nanoleakage [33]. These fibers are more susceptible to hydrolytic degradation caused by activated endogenous enzymes, especially MMP2, MMP9 and cysteine cathepsins, which are native constituents of the fibrillar network of human dentinal organic matrix [34,35]. Included in the matrix as inactivated forms, these pro-enzymes are activated by dentinal acid etching, either by the primers of SE adhesives [36] or by orthophosphoric acid during ER protocol. It has been demonstrated that the consequent degradation of collagen fibrils has an important relevance on the reduction in bond strength over time, with values from 38% to 42.6% at T12 [37,38]. However, the use of the SE approach in addition to a mild pH adhesive may lead to a minor activation of MMPs, being therefore less susceptible to bond strength degradation over time. In fact, it seems evident that generally higher levels of MMPs activity seen in etch-and-rinse adhesives compared with self-etching adhesives seems to correlate with the more rapid destruction of hybrid layers seen in etch-and-rinse bonds, relative to self-etch adhesives [39,40]. Even though no significant difference was found at T12, interesting results on 10-MDP molecule stability were shown by Aida et al., who found superior durability on bond strength over time in adhesives containing 10-MDP over HEMA-only adhesives [41].

These results were also confirmed by a systematic review by Carilho et al. [42] which confirmed that the use of an adhesive system with a hydrophilic spacer carbon chain induces more water sorption and better dentin wettability whereas more hydrophobic functional monomers (MDP) are more suitable in order to avoid the effects of hydrolytic degradation [43,44]. However, the present study results may be partially related to the short observation period (1 year), but it is encouraging to see how, even if with significant differences in bond strength, the bond remains stable over time in all three cements tested.

## 5. Conclusions

Within the limitations of this in vitro study, Panavia V5 seems to have better µTBS values at T = 0 than QM and ES, while ageing did not affect the bond strength of tested systems. These results seem encouraging as aging time did not influence µTBS of adhesive cement tested at 1-year observation interval, which led to speculate a good bond strength stability when luting indirect CAD-CAM restorations. However, further studies are nec-

essary to evaluate this positive trend at a longer observation interval, with implemented adhesive strategies and different CAD-CAM materials.

**Author Contributions:** Methodology, V.E.A., B.A., F.G.; Data Analysis, B.A., C.A.; Software, I.E.; Investigation, F.R., Writing—original draft preparation, V.E.A., C.A.; Writing—review and editing, F.F., S.N.; Conceptualization, C.A., S.N.; Supervision, S.N. All authors have read and agreed to the published version of the manuscript.

**Funding:** This research received no external funding.

**Institutional Review Board Statement:** Not applicable.

**Informed Consent Statement:** Informed consent was obtained from all subjects involved in the study.

**Data Availability Statement:** The data presented in this study are available on request from the corresponding author.

**Conflicts of Interest:** The authors declare no conflict of interest.

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
