# Peer review of "Bond Strength Stability of Different Dual-Curing Adhesive Cements towards CAD-CAM Resin Nanoceramic: An In Vitro Study"

_applsci, doi:10.3390/app11093971_

Round 1

Reviewer 1 Report

The data in this paper is one of the reasons for choosing cement. In the future, it will be more useful data if we consider the effects of occlusal force and thermal recycling.

Author Response

The data in this paper is one of the reasons for choosing cement. In the future, it will be more useful data if we consider the effects of occlusal force and thermal recycling.

Author response: thanks for your suggestion, next studies could be focused on different ageing strategies. The artificial ageing with immersion in artificial saliva is strongly tested in relevant literature on this topic.

Reviewer 2 Report

very interesting and innovative

Author Response

Author response: Thanks for your comment

Reviewer 3 Report

Strengths

In this work 36 molars crowns were flattened in order to expose sound dentin and a standardized smear layer was created with 600 grit paper.  Specimens were divided into six groups according to the luting cement employed (n=12 each): G1:  Panavia V5 (Kuraray, Japan); G2: Bifix QM (Voco, Germany); G3: Estecem (Tokuyama, Japan). CAD-CAM blocks (Cerasmart, GC), shade A2LT, size 14, were sectioned with a diamond saw to obtain 4 mm height specimens, which were then luted on the coronal dentin, following the manufacturer instructions. Specimens were serially sectioned to obtain 1mm thick beams in accordance with the μTBS test technique. Half of the beams were stressed to failure after 24 h (t=0), while the other half was stored in artificial saliva for 12 months, at 37 °C, for ageing before stressing to failure (t=12).  Within the limitations of this study, Panavia V5 seems to have better μTBS values 267 at T=0 than QM and ES, while ageing did not affect the bond strength of tested systems.

Weaknesses

1. Unfortunately, there is no Figure 5. (Line 172).

2. The discussion of the results is quite complex, so it is necessary to supplement it with the relevant conclusions.

3. References need to be given the most recent. The authors do not have enough new references. Make references in line with the requirements of the journal.

Author Response

  1. Unfortunately, there is no Figure 5. (Line 172). 

Authors response: thanks for the comment, figure 5 has been added to the manuscript

2. The discussion of the results is quite complex, so it is necessary to supplement it with the relevant conclusions.

Authors response: thanks for the comment, the conclusions has been improved to better clarify the results

3. References need to be given the most recent. The authors do not have enough new references. Make references in line with the requirements of the journal.

Authors response: thanks for the comment, references were updated